# Non-Parametric Inference of Relational Dependence

**Ragib Ahsan**[1]        **Zahra Fatemi**[1]        **David Arbour**[2]        **Elena Zheleva**[1]

[1]Department of Computer Science, University of Illinois at Chicago, Chicago, IL, USA
[2]Adobe Research, USA

## Abstract

Independence testing plays a central role in statistical and causal inference from observational data. Standard independence tests assume that the data samples are independent and identically distributed (i.i.d.) but that assumption is violated in many real-world datasets and applications centered on relational systems. This work examines the problem of estimating independence in data drawn from relational systems by defining sufficient representations for the sets of observations influencing individual instances. Specifically, we define marginal and conditional independence tests for relational data by considering the kernel mean embedding as a flexible aggregation function for relational variables. We propose a consistent, non-parametric, scalable kernel test to operationalize the relational independence test for non-i.i.d. observational data under a set of structural assumptions. We empirically evaluate our proposed method on a variety of synthetic and semi-synthetic networks and demonstrate its effectiveness compared to state-of-the-art kernel-based independence tests.

## 1 INTRODUCTION

Measuring dependence is a fundamental task in statistics. However, most existing independence tests assume that the observed data is independent and identically distributed (i.i.d.). This assumption makes them unsuitable for capturing statistical dependencies in real-world relational systems, from social networks to protein-protein interactions, in which data instances depend on each other.

Relational dependence refers to a statistical dependence, marginal or conditional, between random variables in which at least one of the variables is a relational variable [Lee and Honavar, 2017]. A relational variable is a set of random variables that belong to instances related to an instance of interest, such as friends of a person or proteins interacting with a target protein. Relational dependence testing is central to social influence studies and causal discovery in relational systems, yet there is no standard statistical tool for inferring different forms of dependence from observational relational data. In this paper, we present a practical tool for determining marginal and conditional independence between relational variables with consistency guarantees.

To understand the challenge of estimating relational dependence, let's consider the following example:

**Example 1.** *Sally doesn't smoke but she has some friends who smoke. Sally starts to smoke over time. Are the smoking habits of Sally's friends and Sally's decision to take up smoking independent?*

Part of the problem of detecting this dependence is that we need to know a priori the exact mechanism of dependence. Is it because all of Sally's friends share a certain behavior? Or is there a minimum threshold of friends necessary (e.g., half of her friends smoke) to activate the dependence? The nature of the relationships among her friends might play a role as well. Without prior knowledge of this mechanism of dependence, any existing statistical test is likely to fail. As such, the focus of our work is to develop a flexible non-parametric test that can capture multiple forms of dependence. Note that for the scope of this work, we do not focus on the causal aspects of this question (e.g., social influence) but the test that we propose is applicable to causal discovery as well.

Recent studies have proposed non-parametric independence tests designed for non-i.i.d data. [Flaxman et al., 2015, Lee and Honavar, 2017]. Flaxman et al. [2015] develop a test between propositional variables accounting for latent homophily in a grid network but do not consider relational variables. Lee and Honavar [2017] develop a test for conditional independence (KRCIT) between relational variables which is the current state-of-the-art test for relational dependence. They operationalize the test by flattening the relational data

*Accepted for the 38th Conference on Uncertainty in Artificial Intelligence* (UAI 2022).

into a single, propositional table. However, the current state-of-the-art test has three key limitations. First, it requires practitioners to make explicit assumptions about the data generating processes and to specify an aggregation function over the relational variable a priori. Second, existing tests rely on propositionalization, which refers to the process of projecting connected data to a single, propositional table, which raises statistical concerns Maier et al. [2013b]. Third, it is computationally expensive and inapplicable to large relational datasets.

In this work, we focus on developing a general definition of relational dependence and a statistical test, NIRD (Non-parametric inference of relational dependence), that is able to capture a family of aggregate functions for characterizing relational dependence. The contributions of our work are in providing 1) complete definitions of marginal and conditional relational dependence which extend the classical definition of Daudin [1980], 2) independence tests with consistency guarantees, 3) the explicit representation of neighboring sets through kernel mean embeddings, and 4) a scalable test through random Fourier feature formulation which makes the test practical for real-world applications. We compare our proposed test to KRCIT on a variety of synthetic networks and simulate several social network characteristics, such as structure, density, and size. We demonstrate the applicability of our method for detecting peer influence using both semi-synthetic and real-world social networks.

## 2 RELATED WORK

There are several independence tests from observational data. Partial correlation is used for the independence test on Gaussian variables with linear dependence [Baba et al., 2004]. A more advanced test, the Hilbert-Schmidt independence criterion (HSIC) [Gretton et al., 2005] is a statistical test that has been extended to marginal independence testing for structured data [Zhang et al., 2009] and random processes [Chwialkowski et al., 2014]. Flaxman et al. [2015] utilize HSIC to develop marginal and conditional independence tests for propositional variables in the presence of a latent confounder in a single-entity network with an additive noise generating function. Relational dependence exhibits itself through a pairwise edge in the network which is a noisy surrogate of latent homophily. Lee and Honavar [2017] extend these tests to explicitly include relational variables. They operationalize the tests by considering pairwise dependence between each member of the relational variable set in the flattened data. However, existing tests require prior domain knowledge in order to detect complex dependencies between relational variables such as entropy, variance or reaching a threshold of peers' actions by requiring the specification of aggregation functions. We lessen this requirement by employing kernel mean embeddings, which provide a non-parametric aggregation capable of capturing a large

number of aspects of smooth functions over distributions belonging to the exponential family.

Relational independence testing is central to causal discovery and structure learning of relational models. However, existing relational causal discovery algorithms either rely on the existence of a relational independence oracle [Maier et al., 2010, 2013a, Lee and Honavar, 2016] or use conditional independence tests developed for i.i.d. data [Lee and Honavar, 2019], making them less practical for real-world applications. The relational dependence tests in this work together with existing tests for detecting causal direction of relational dependence [Arbour et al., 2016b] can make relational causal discovery achievable for real-world datasets with unknown dependence functions.

One of the applications of relational dependence testing is in characterizing social influence in observational relational data. Existing social influence studies measure influence through predictive models [Christakis and Fowler, 2007, Bakshy et al., 2011], or randomized controlled trials [Muchnik et al., 2013, Su et al., 2020]. A standard statistical tool to infer different forms of social influence from observational relational data does not exist. Our work can help develop such tools by treating social influence detection as an instance of a relational independence test.

## 3 PROBLEM DEFINITION

The central task of this paper is to develop a statistical test for determining marginal and conditional independence between relational variables. Before formalizing the problem, we introduce the necessary notations and definitions. We denote random variables and their realizations with uppercase and lowercase letters, respectively, and bold to denote sets. We use an Entity-Relationship model [Heckerman et al., 2007] to describe relational data following previous work [Maier et al., 2010, Lee and Honavar, 2017]. A relational schema $\mathcal{S} = \langle \mathcal{E}, \mathcal{R}, \mathcal{A} \rangle$ represents a relational domain where $\mathcal{E}$, $\mathcal{R}$ and $\mathcal{A}$ refer to the set of entity, relationship and attribute classes respectively. We consider an undirected graph $G$ to be the instantiation of the relational schema $\mathcal{S}$ where nodes and edges refer to the entity and relationship instances respectively. A given schema can entail numerous possible instantiations. For ease of exposition, we focus on a schema with a single entity type (e.g., Person) and a single relationship type (e.g., Friends) but discuss the extensions necessary to address multi-relational data in the Appendix. We refer to the set of vertices, edges and adjacency matrix of $G$ as $\boldsymbol{V}$, $\boldsymbol{E}$ and $A$ respectively. For example, nodes $v_1 = Ana$ and $v_2 = Bob$ with undirected edge between them can come from the $Person - Friends - Person$ relations. The attribute class $\mathcal{A}$ represents the set of possible attributes for the specified entity of the given graph $G$. Let $v_i.X$ and $v_i.x$ refer to the attribute $X \in \mathcal{A}$ and its realization respectively for node $v_i \in \boldsymbol{V}$ corresponding

to instance $i \in \mathcal{E} \cup \mathcal{R}$. Here, $v_i.X$ is considered to be a propositional random variable. Following prior work we define a relational variable as a set of propositional random variables [Maier et al., 2013a, Lee and Honavar, 2017].

**Definition 1** (Relational Variable). *Given a relational schema $\mathcal{S} = \langle \mathcal{E}, \mathcal{R}, \mathcal{A} \rangle$, its instantiation $G$ and a path predicate $\rho$, a relational variable $\sigma(v_i, \mathbf{X}, G, \rho)$ is the set of attributes $v_j.\mathbf{X}$ selected by $\rho$ of nodes $v_j \in \mathbf{V}$ reachable from $v_i \in \mathbf{V}$ such that $\mathbf{X} \subseteq \mathcal{A}$, where the path predicate $\rho$ is a function given by: $\rho(v_i, G) : \mathbf{V} \mapsto \mathcal{P}(\mathbf{V})$.*

Here, $\mathcal{P}(\mathbf{V})$ refers to the power set of $\mathbf{V}$. An example path predicate is $\rho(v_i, G) = \{v_j | v_j \in \hat{\mathcal{N}}(v_i)\}$ where $\hat{\mathcal{N}}(v_i)$ refers to the direct neighbors of $v_i$ in $G$ and the corresponding relational variable $\sigma(v_i, \mathbf{X}, G, \rho)$ refers to the set of attributes of the neighboring nodes. For simplicity and assuming only direct neighbors throughout the paper, we denote the relational variable corresponding to attribute $X$ by $\sigma_X(v_i)$ and its value by $\sigma_x^{v_i}$. Note that $\sigma_X(v_i)$ can represent a *propositional* variable as a special case. For example, $\sigma_X(v_i) = \{v_i.X\}$ refers to the $X$ attribute of $v_i$. We also make the following assumptions:

**A 1.** *Each node $v \in V$ has degree of at least 1.*

**A 2.** *The adjacency matrix of $G$ is symmetric with edge weights bounded by some real constant.*

**A 3.** *Dependence between two instances $i$ and $j$ implies the existence of a path in the graph between $v_i$ and $v_j$.*

Relational dependence refers to a statistical dependence, either marginal or conditional, between two variables where at least one of the variables is relational. The goal of a relational dependence test is to determine whether to reject the null hypothesis of independence between these variables or not. The representation of relational data for such a test is non-trivial because data instances are not i.i.d. A common practice to deal with relational data is *propositionalization* [Kramer et al., 2001], which refers to the process of projecting a set of connected data samples down to a single, propositional table. In the context of relational dependence testing, flattening has three main deficiencies. First, the entities in the flattened data are not i.i.d. Second, choosing the appropriate aggregation function is non-trivial as discussed in section 1. Failing to appropriately define the aggregate in this case could lead to increased type I errors in marginal tests, and both type I and II errors for conditional tests. Third, flattening raises statistical concerns for relational causal discovery, one of the application areas of relational conditional independence tests, by violating the causal Markov condition [Maier et al., 2013b]. Lee and Honavar [2017] address the first deficiency by proposing a solution framework based on graph kernels using an existing i.i.d. kernel-based CI test method. However, their approach does not directly address the other two concerns.

Let's look at the problem with a concrete example. We consider an entity Person which exhibits attributes such as smoking status before ($S0$) and after ($S1$) a given time threshold $t$ and $G$ represents the network of social ties. Detecting the dependence of peers on a person's smoking behavior can be formalized as an independence test. For example, detecting whether one's smoking behaviour is marginally independent of one's direct friends' smoking behaviour could be carried out by a marginal test of $v_i.S1 \perp\!\!\!\perp \sigma_{S0}(v_i)$. Similarly, a conditional test of $v_i.S1 \perp\!\!\!\perp \sigma_{S0}(v_i) | v_i.S0$ should detect whether one's current smoking behavior is independent of friends' old smoking behavior given one's old smoking behavior.

In this work, we propose a relational dependence test which captures complex dependencies between relational variables without relying on flattening or explicit aggregate representations. We extend the definition of conditional independence for non-parametric functions by Daudin [1980] and propose the following definitions of marginal and *relational* conditional independence:

**Definition 2** (Relational Marginal Independence). *Two relational variables, $\sigma_X(v_i)$ and $\sigma_Y(v_i)$ are said to be marginally independent of each other if and only if, $\mathbb{E}\left[g_X(\sigma_X(v_i))g_Y(\sigma_Y(v_i))\right] = \mathbb{E}\left[g_X(\sigma_X(v_i))\right]\mathbb{E}\left[g_Y(\sigma_Y(v_i))\right]$ for any smooth square measurable functions $g_X(\cdot), g_Y(\cdot)$.*

**Definition 3** (Relational Conditional Independence). *Two relational variables, $\sigma_X(v_i)$ and $\sigma_Y(v_i)$ are said to be independent of each other given a third, $\sigma_Z(v_i)$ if and only if, $\mathbb{E}\left[g_X(\sigma_X(v_i))g_Y(\sigma_Y(v_i))|g_Z(\sigma_Z(v_i))\right] = \mathbb{E}\left[g_X(\sigma_X(v_i))|g_Z(\sigma_Z(v_i))\right]\mathbb{E}\left[g_Y(\sigma_Y(v_i))|g_Z(\sigma_Z(v_i))\right]$ for any smooth square measurable functions $g_X(\cdot), g_Y(\cdot), g_Z(\cdot)$.*

Here, $g_X(\cdot), g_Y(\cdot), g_Z(\cdot)$ are *aggregate* functions that map $\sigma$ to a real-valued vector. They could be *sum*, *mean* or any other complex non-linear function. The rejection of the null hypothesis of marginal independence would mean that the variables are possibly dependent, either due to a directed path between them or due to a direct, causal relationship, or the presence of a confounding relationship. For a relational conditional independence (RCI) test, the rejection of the null hypothesis would imply that the two variables are not independent given the conditioning set. Note that because we are considering the dependence between sets of relational variables and their propositional counterparts we circumvent the three problems with flattening described earlier.

# 4 RELATIONAL INDEPENDENCE TESTS

In this section, we discuss the components which operationalize the definition of relational dependence into an empirical test. We first describe a non-parametric relational aggregate formed by local kernel means. Then we formulate marginal and conditional independence tests using the

kernel mean embedding. Then, we discuss the theoretical boundaries for the consistency of the proposed test. Finally, we introduce techniques for large-scale approximation of the proposed relational kernels that can speed up the independence test significantly.

## 4.1 NON-PARAMETRIC AGGREGATE REPRESENTATIONS

One of the central problems in estimating dependence in relational settings is defining a sufficient representation for the sets of observations for individual instances of a relational variable. Prior work [Maier et al., 2013a, Arbour et al., 2016a, Lee and Honavar, 2017] considered aggregation functions, usually one, which are specified *apriori* by the practitioner. However, in many scenarios, it is unreasonable to expect practitioners to reason over a very complex joint distribution or to know the exact parametric form of dependence. For example, the possible aggregation in effect for the spread of obesity in social networks [Christakis and Fowler, 2007] can be different from people's influence on the Twitter platform [Bakshy et al., 2011]. A generalized definition and associated operationalization of relational dependence can help the practitioner by directly measuring dependence without prior domain knowledge about aggregations on the given relational system.

The distance between the embedding of joint distribution and embedding of product of marginals can be used to infer independence according to definition 2, while avoiding explicit density estimation as an intermediate step. Note that the aggregate functions are represented implicitly through the kernel mean embedding (KME) [Smola et al., 2007, Muandet et al., 2017]. An appealing property of the kernel mean embedding is that if the kernel is universal then the kernel mean uniquely represents all moments for any member of the exponential family [Smola et al., 2007] [1].

Adopting the kernel mean as an aggregation function removes the burden of reasoning over parametric families and predefined aggregates. Specifically, the kernel mean embedding considers the mean of a variable after applying a projection $\phi(\cdot)$ into some RKHS, $\mu = \int \phi(x)p(x)dx$, with the corresponding empirical estimate of $\hat{\mu} = \frac{1}{N}\sum_i^N \phi(x_i)$ where $N$ is the number of observations and $x_1, \ldots, x_N$ are observations from a random variable $X$ [Smola et al., 2007].

We present the practical implementation of the kernel mean as a relational aggregate. For a given node $v_i$, we define the kernel mean aggregate of its neighbors with respect to the attribute $X$ as $\mu(v_i) = \frac{1}{\deg(v_i)}\sum_{m\in\hat{\mathcal{N}}(v_i)} \phi(m.x)$

where $\hat{\mathcal{N}}(\cdot)$ refers to a path predicate which is restricted to

immediate neighbors for ease of exposition. Because $\phi$ may map to an infinite dimension, it is impractical to explicitly represent this quantity. Fortunately, because our statistics of interest are concerned with the covariance, the kernel trick, i.e. considering the inner product rather than the feature representations directly, can be employed. Specifically, the inner product between relational kernel mean is given as

$$\langle \mu(v_i), \mu(v_j) \rangle = \frac{\sum_{m\in\hat{\mathcal{N}}(v_i)} \sum_{p\in\hat{\mathcal{N}}(v_j)} k(m.x, p.x)}{\deg(v_i)\deg(v_j)},$$

which can be written for an entire sample in terms of a matrix product between the network adjacency matrix, $A$, the inverse degree matrix $D^{-1}$ where $D_{i,i} = \frac{1}{\deg(v_i)}$, and the kernel matrix $K_X$, by observing $(D^{-1}A\phi(\mathbf{x}))(D^{-1}A\phi(\mathbf{x}))^T = D^{-1}AK_XAD^{-1}$.

In contrast to the propositional kernel mean, the convergence of the relational to its population counterpart is not necessarily guaranteed because of sample dependence. We discuss convergence and consistency guarantees under the assumption of weak dependence after describing the relational independence tests.

## 4.2 RELATIONAL MARGINAL INDEPENDENCE TEST

With the relational kernel mean defined we now turn to the central task of this paper, non-parametric inference of relational dependence (NIRD). As a test statistic, we use the Hilbert-Schmidt independence criterion (HSIC) [Gretton et al., 2005]. HSIC measures the maximum distance between an embedding of the observed joint distribution, and the product of the marginals, i.e., $\|\mathbb{E}[\phi(x) \otimes \phi(y)] - \mathbb{E}[\phi(x)] \otimes \mathbb{E}[\phi(y)]\|^2$. We perform a hypothesis test using HSIC as the test statistic where the null hypothesis refers to independence. The test produces a p-value which is used to decide whether to reject the null or not. Testing relational independence using HSIC is straightforward with the relational kernel mean by using the kernel matrix defined earlier in the empirical HSIC estimator. Defining the centering matrix $H = I - \frac{1}{n}\mathbf{1}\mathbf{1}^\top$, an empirical estimate of HSIC is given by $\frac{1}{n^2}\text{trace}(K_XHK_YH)$, where $K_X$ and $K_Y$ are kernel matrices corresponding to the random variables $X$ and $Y$, respectively. Independence testing with HSIC can be performed by using the corresponding relational kernel in the test statistic.

## 4.3 RELATIONAL CONDITIONAL INDEPENDENCE TEST

A similar construction can be employed to test for relational conditional independence, defined in Definition 3. Following Strobl et al. [2019], we consider the following

---

[1] We refer readers to Szabó and Sriperumbudur [2017] for conditions for a kernel to be universal. Many popular kernels such as the RBF kernel are universal

$L^2$ spaces,

$$F_{XZ} \triangleq \left\{ \tilde{f} \in L^2_{XZ} \mid E(\tilde{f} \mid Z) = 0 \right\}$$

$$F_{YZ} \triangleq \left\{ \tilde{g} \in L^2_{YZ} \mid E(\tilde{g} \mid Z) = 0 \right\}$$

$$F_{Y \cdot Z} \triangleq \left\{ \tilde{h}' \mid \tilde{h}' = h'(Y) - E\left(h' \mid Z\right), h' \in L^2_Y \right\}$$

Each of these quantities can easily be constructed by considering regressions, e.g. $\tilde{f}$ can be obtained by taking the residuals after performing a regression. We consider a mean of the feature basis representation as an aggregation function whenever one of the variables is relational. Under the assumption that the direct sum of the reproducing kernel Hilbert spaces, $k_x k_y$ and $k_z$ is dense in $L_2$, Strobl et al. [2019] (proposition 5) showed that conditional linear covariance of zero implies uncorrelatedness, i.e., $\mathbb{E}\left[\tilde{f}\tilde{g}\right] = 0 \implies X \perp\!\!\!\perp Y \mid Z \implies \Sigma_{XY|Z} = 0$. This motivates the use of a multiple output kernel ridge regression as an estimator of the conditional expectation, $\beta = (\phi(z)^T\phi(z) + \lambda I)^{-1}\phi(z)^T\phi(\ddot{x})$ where $\ddot{X} \triangleq (X, Z)$ is the concatenation of x and z. Informally, this can be seen as applying the "kernel trick" of considering linear operations on non-linear transformations of the data allowing for observations to be dependent. The test is then constructed by considering the residuals, $\widetilde{\phi(\ddot{x})} = \phi(\ddot{x}) - \phi(z)\beta_{\ddot{x}z}, \widetilde{\phi(y)} = \phi(y) - \phi(z)\beta_{yz}$ and sum of the squared covariances between them. The final form of the test is given by $\frac{1}{n^2}\text{trace}\left(\widetilde{K_{\ddot{X}}}H\widetilde{K_Y}H\right)$ where $\widetilde{K_{\ddot{X}}}$ and $\widetilde{K_Y}$ refers to the kernel matrices for the residuals $\widetilde{\phi(\ddot{x})}$ and $\widetilde{\phi(y)}$ respectively.

There are two considerations in employing this procedure in a relational setting, namely how to handle relational variables in the conditioning set ($Z$) and the test set ($X$), respectively. When a member of the conditioning set is relational, the test procedure is identical after replacing $\phi(z)$ with its relational counterpart, $\frac{1}{|\hat{\mathcal{N}}(z)|}\sum_{m \in \hat{\mathcal{N}}(z)} \phi(m)$. When a member of the test set is relational, the problem is reduced to predicting each member of the set independently by considering the regression of the perspective of the relational variable, as described by Maier et al. [2013a]. After regressing individual members, the mean of residuals is then considered for the marginal tests, $\widetilde{\sigma(\phi(x))} = \frac{1}{|\hat{\mathcal{N}}(x)|}\sum_{m \in \hat{\mathcal{N}}(x)} \phi(m) - \phi(z)\beta_{mz}$.

## 4.4 CONSISTENCY OF RELATIONAL INDEPENDENCE TEST

In order to reason about the behavior of test statistics under non-i.i.d. samples and understand asymptotic behavior we need to characterize the behavior of dependence amongst instances as a function of some notion of distance between instances. There are a number of formalisms for reasoning about dependent data [Andrews and Pollard, 1994, Bickel

and Bühlmann, 1999, Dedecker et al., 2007]. In this work we focus on weak dependence [Dedecker et al., 2007], which we describe next.

### 4.4.1 Weak Dependence

In order to accommodate dependent observations and maintain consistency of the testing procedure we will assume that observations are weakly dependent. Weak dependence provides a flexible notion of dependence that requires only the definition of distance between instances and the presence of a measurable probability space. Within this work we will make use of the notion of weak dependence, i.e. $\tau$-dependence.

**Definition 4.** *[Dedecker et al., 2007] Let $\boldsymbol{\pi}$ be a filtration[2] over the set of nodes in a graph, G, defined by performing a breadth first search at an arbitrary node, $v \in G$. Further, define $X$ to be a $\mathbb{L}^p$-integrable random variable. The **weak-dependence coefficient** is defined as $\tau_{p,r}(X) = \sup_{(i,j)} \| \sup_g Cov(g\left(X_{\boldsymbol{\pi}(i)}\right), g\left(X_{\boldsymbol{\pi}(j)}\right))\|_p$, where $i \leq j$ and $j - i \leq r$, and $g()$ is a Lipschitz function.*

Intuitively, the weak dependence coefficient, $\tau_{p,r}(X)$ measures the covariance between a vector, $X_i$ and another random vector $X_j$ drawn from the same process separated by at least distance of $r$. We call a process *weakly dependent* if $\tau$ tends to zero as the $r$ tends to infinity. Note that this is a strictly weaker condition than alternative assumptions on dependence such as strong mixing and $m$-dependence which require independence at a finite distance, whereas weak-dependence only requires it asymptotically.

### 4.4.2 Weak Dependence in Relational Domains

We provide a natural extension of weak dependence within the relational setting by replacing usual definition of distance to the shortest path distance between two nodes in a graph. The role of $\tau$ in this case can be interpreted as measuring the decay of dependence between instances as a function of shortest-path distance. We will assume from here out that as the distance between any two nodes in the network tends to infinity, the dependence between them converges to zero. More formally, we will employ the following assumption:

**A 4.** *$(X_t)_{t \in \pi}$ is a strictly stationary $\tau$-dependent process with $\sum_{r=1}^{\infty} r^2\sqrt{\tau_r(X)} \leq \infty$ for some filtration $\pi$, where $r$ is shortest-path graph distance.*

The notion of weak dependence within the network setting is not novel to this work, Xiang and Neville [2011] make use of the $\tau$-coefficient in the context of deriving asymptotic

---

[2] A filtration is an ordering of a set such that for any two subsets, $S_{1,\ldots,j}, S_{1,\ldots,k}, j \leq k \to S_{1,\ldots,j} \subseteq S_{1,\ldots,k}$.

consistency for transductive learning with an assumption of linear dependence amongst instances. However, to our knowledge, our work is the first to consider weak dependence with arbitrary dependence for independence testing of relational data. Consistency of the relational independence testing is provided by the following theorem and corollary, after applying two additional assumptions.

**A 5.** *The maximum degree of any node in the network is bounded by a real constant.*

**A 6.** *The network structure is fixed and doesn't change during the generation of the observed random variables.*

Assumption 5 ensures that the average shortest path distance from any node to all other nodes in the graph tends to infinite as the number of nodes tends to infinite, which is necessary in order to have convergence of weakly dependent sequences. Assumption 6 ensures that the observed neighborhoods for nodes correspond to the structure which generated the data.

**Theorem 1.** *Under the aforementioned assumptions the Hilbert-Schmidt independence criterion of two weakly dependent propositional variables converges in $L_1$ to its population counterpart, i.e., $|HSIC_n - HSIC_{population}| \xrightarrow{d} 0$.*

**Corollary 1.** *Under the aforementioned assumptions the Hilbert-Schmidt independence criterion between a weakly relational and a weakly dependent propositional variable converges in $L_1$ to its population counterpart, i.e., $\left|\widehat{HSIC_n} - HSIC_{population}\right| \xrightarrow{d} 0$.*

The proof of theorem 1, which is deferred to the supplement, follows by observing that the empirical estimate of HSIC is a degenerate $V$-statistic and then through a proof which shows consistency of degenerate $V$-statistics under weak-dependence in structured domains, which may be of independent interest. Similarly, the proof of the corollary, also deferred to the supplement follows from theorem 1 and showing that the weak dependence coefficient remains finite for relational variables.

It is important to note that Theorem 1 and Corollary 1 show convergence in distribution but do not claim any guarantees regarding the rates of convergence with respect to the number of nodes and level of dependence. The rate of convergence will depend on the weak dependence coefficient. In the case that the coefficient is 0, this reduces to results that correspond to prior work on iid data [Zhang et al., 2011]. While there is prior work studying this in more restrictive assumptions on the dependence between instances [London et al., 2013], we are not aware of similar results for the case of weak dependence in general structured domains even in the simpler case of regression. This would be an important direction for future work.

## 4.5 LARGE SCALE APPROXIMATIONS

While the proposed model is theoretically appealing, the associated time and space complexity render it infeasible for most modern network settings. To address this, we appeal to an approximation of the kernels known as Random Fourier Features [Rahimi and Recht, 2008]. Random Fourier Features exploit Bochner's theorem, which states that a continuous, time-invariant kernel is positive definite if and only if the kernel is the Fourier transform of some non-negative measure. For example the Gaussian kernel can be represented with the following Fourier transformation $\hat{k}(\omega) = \frac{1}{2\pi} \int e^{-j\omega^\top \delta} k(\delta) d\delta$. This property implies that a kernel can be approximated via the following procedure:

- Draw $D$ samples, from some distribution (i.e Normal), to approximate the Gaussian kernel where the variance $\sigma$ corresponds to the bandwidth of the kernel.
- Construct the Fourier basis explicitly as $z(x) = \sqrt{\frac{2}{d}} \left[\cos\left(w_1^T x\right), \sin\left(w_1^T x\right), \ldots, \right]$.
- Perform linear operations using $z$.

Following [Zhang et al., 2018, Strobl et al., 2019], we approximate HSIC using random Fourier features by considering $\widehat{HSIC}(X, Y) = \left\|\frac{1}{n} \boldsymbol{Z}_X^T \boldsymbol{H} \boldsymbol{Z}_Y\right\|^2$ where $Z$ is a $n \times d$ dimensional matrix with each row consisting of the random Fourier features for an observation. We can represent the relational kernel mean as $D^{-1}AZ$, and the corresponding test statistic as $\left\|\frac{1}{n} Z_X^T A D^{-1} H Z_Y\right\|$ where $D$ and $A$ are the diagonal degree and adjacency matrix as before. In several experiments we show that using approximate statistic leads to significant performance improvements with minimal effect on the efficacy of the test, even with only a few random features.

## 5 EXPERIMENTS

We run experiments with multiple network datasets, relational dependence cases, and synthetic attribute generators to evaluate the effectiveness of the proposed test.

## 5.1 NETWORK DATASETS

We consider networks from two synthetic graph generators and three non-PII real-world networks. First, for the Barabási-Albert (BA) model, we vary the parameter that controls the number of nodes a new node can attach to. For the Erdős-Rényi (ER) model, we vary the probability of edge creation between each pair of nodes. For each set of parameters, we generate 100 networks with size 100. The small size of the synthetic networks is driven by the baseline method which does not scale well, as shown in Figure 4a. We also demonstrate the applicability of our approach through a Facebook ego-network with $4,039$ nodes and

88, 234 edges [Leskovec and Mcauley, 2012]. The other two real-world datasets (*Twitter*, *50 Women*) and corresponding experimental results are described in the Appendix.

## 5.2 FOUR CASES OF RELATIONAL (IN)DEPENDENCE

We choose three representative relational dependence cases and one relational independence case to cover a range of possible tests. We consider attributes $Z, X, Y \in \mathcal{A}$ which measure characteristics in time steps $t-1, t, t+1$ respectively. All the cases are represented with arrows showing the direction of dependence:

1. **Case 1:** $\sigma_X(v_i) \rightarrow v_i.Y$
2. **Case 2:** $\sigma_X(v_i) \leftarrow v_i.Z \rightarrow v_i.Y \leftarrow \sigma_X(v_i)$
3. **Case 3:** $v_i.X \leftarrow \sigma_Z(v_i) \rightarrow v_i.Y \leftarrow v_i.X$
4. **Case 4:** $\sigma_X(v_i) \leftarrow v_i.Z \rightarrow v_i.Y$

where $\sigma_X(v_i)$ and $\sigma_Z(v_i)$ are relational variables on the attributes $X$ and $Z$ of the direct neighbors of $v_i$. Case 1 refers to marginal independence between a relational and a propositional variable ($\sigma_X(v_i) \perp\!\!\!\perp v_i.Y$). Cases 2 and 3 introduce conditional independence given a confounder. Case 2 refers to a propositional confounder ($\sigma_X(v_i) \perp\!\!\!\perp v_i.Y | v_i.Z$) whereas case 3 refers to a relational confounder ($v_i.X \perp\!\!\!\perp v_i.Y | \sigma_Z(v_i)$). A test should be able to reject the null hypothesis of no dependence in the first three cases. Case 4 represents conditional independence and the test should not reject the null hypothesis and it should produce high errors. Note that direction is ignored in the test. The synthetic attribute generation process is described in Appendix.

## 5.3 EXPERIMENTAL SETUP

We empirically evaluate the proposed approach, NIRD, to the state-of-the-art RCI test method, KRCIT [Lee and Honavar, 2017]. [3] We report the average Type I and Type II errors with significance level 0.05 over 100 trials for each set of parameters. We use *Radial Basis Function kernel* (RBF) as the base kernel. KRCIT is implemented with HSIC as the kernel-based marginal independence test method and KCIT [Zhang et al., 2011] as the kernel-based conditional independence (CI) test. We use the approximate method of NIRD in all experimental evaluation with 20 and 50 random Fourier features for marginal and conditional test respectively. We estimate the null distribution via permutation on the non-relational variable since the marginal distribution remains unchanged. We compare both RCI methods (NIRD, KRCIT) to a recent i.i.d. CI test method, Sobolov Independence Criterion (SIC) [Mroueh et al., 2019] (see

---

[3] Code available at https://github.com/edgeslab/nird-uai22

---

Appendix). We study NIRD's strengths and weaknesses in five experimental setups:

**Relational dependence sensitivity**: We evaluate the sensitivity of the dependence tests to different relational dependence strengths. We report results on polynomial models while varying the dependence coefficient in range $\{0.1, 0.3, 0.5, 0.7, 0.9\}$ for the alternate hypothesis and report both Type I, II errors. Edge connectivity is 3 for BA and edge probability is 0.02 for ER model.

**Diffusion**: We apply NIRD to test for contagion by simulating a linear threshold model [Granovetter, 1978] with initial treatment probability of 0.1 on the semi-synthetic Facebook network. We reassign treatment values in each diffusion step and generate outcomes ($Y$) based on treatments ($X$) generated in the last diffusion step. The attribute generation process is described in the Appendix. We expect the distribution of $\sigma_X(v_i)$ to change with increasing diffusion steps and investigate at what step it is possible to detect relational dependence. We vary the number of steps, sample size and measure Type II error. We also investigate the impact of activation probability on the Type-II error on the Twitter ego-network with 10,000 nodes (results in the Appendix).

**Network sensitivity**: We examine performance over a variety of network structures. We vary edge connectivity of BA in range $\{1, 2, 3\}$ and edge probability of ER in range $\{0.005, 0.01, 0.015, 0.02, 0.025\}$. We use a fixed dependence coefficient value of 0.5.

**Scalability**: To compare the scalability of NIRD against the baseline KRCIT, we generate Erdős-Rényi synthetic networks (edge probability 0.02) with varying number of nodes and report their execution time for marginal and conditional independence testing. We vary the network size (x-axis) in the range $\{100, 200, 300, 400, 500\}$. The choice of small networks was driven by the fact that KRCIT scales exponentially as shown in Figure 4a and it is impossible to run for larger networks. In the Appendix, we also demonstrate the scalability of NIRD through the diffusion experiment which is run on networks of size up to 10k nodes. The experiment is run on a 2.4GHz 8-core machine with 50GB memory.

**Real world demonstration**: In the Appendix, we demonstrate the applicability of our test for detecting peer influence in a well-studied real-world social network (*50 Women*) where our test discovers smoking-, drug- and sport-related peer dependencies that concur with previous research.

## 5.4 RESULTS

**Relational dependence sensitivity**. Figure 1 shows Type I and Type II errors for the polynomial dependency model on synthetic data. The rows correspond to the network models and the columns to relational dependence cases. The solid and dashed lines correspond to Type II and Type I errors

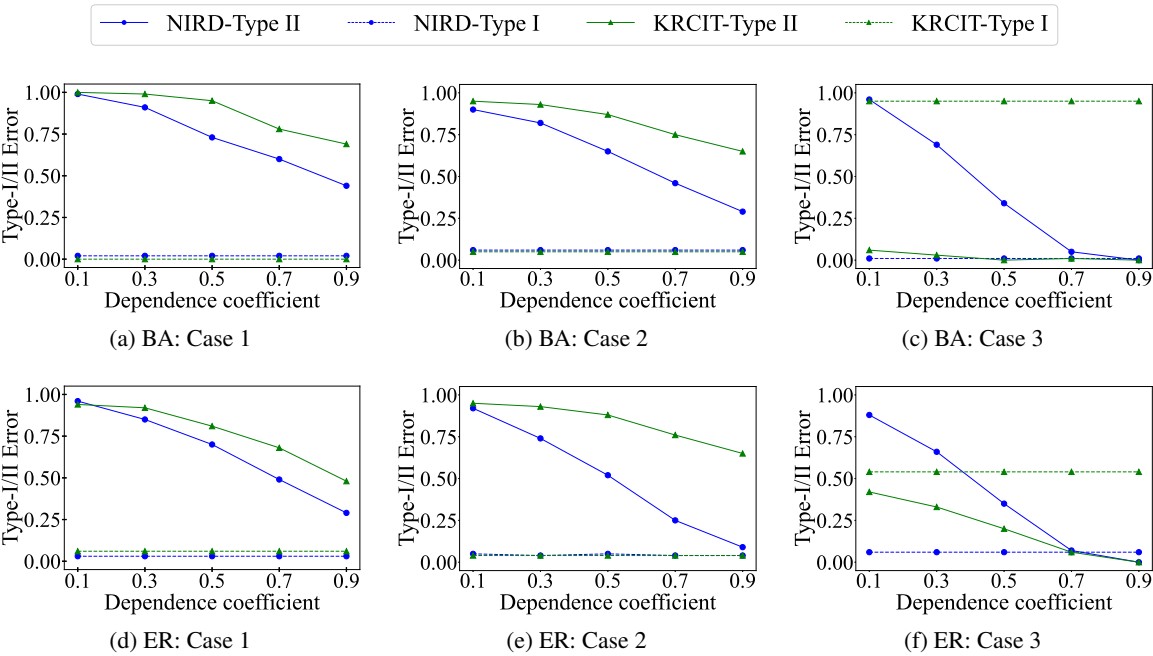

Figure 1: Relational dependence impact on Type I/II errors.

respectively. The test is most challenging when the dependence coefficient, $\beta_d$ (x-axis) is low. The figure shows that both RCI methods are well calibrated with low Type I error (max 0.06 by KRCIT in Erdős-Rényi) for the first two cases. In these cases, NIRD consistently produces lower Type II errors compared to KRCIT. It is most visible in Erdős-Rényi model (1e) with $86\%$ reduction in Type II error for $\beta_d = 0.9$. The performance gain of NIRD increases slightly from case 1 to case 2 as the difficulty increases. In case 3, KRCIT is poorly calibrated and exhibits an unusually high Type I error. Across cases, NIRD shows desired behavior: it is consistently well-calibrated and its Type II error decreases with the increase of relational dependence. Case 4 provides a sanity check and both methods produce high Type II errors (0.9 to 1.0) with good calibration. The error is nearly constant irrespective of strength of dependence coefficients or network model parameters used. In order to test for sensitivity to noise, we repeat these experiments varying the noise variance over multiple trials instead of drawing from a fixed distribution. The results look very similar (see Appendix).

**Diffusion**. Figure 2a shows the impact of the number of diffusion steps (lines) and sample size (x-axis) on the effectiveness of NIRD. At initial activation (1 diffusion step) there is a high Type II error across sample sizes which decreases with higher number of steps. We see a significant decrease in error with just 5 diffusion steps. Further steps drastically lower the Type II error and at 20 steps and larger samples it can reject the null hypothesis consistently. This suggests that relational dependence is easier to detect after several diffusion steps rather than at early activation. It also demonstrates the effectiveness and scalability of NIRD in

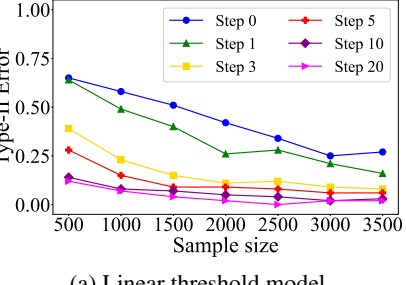

(a) Linear threshold model

Figure 2: Impact of sample size on Type II error.

terms of detecting social phenomena in real world networks. Note that it is computationally infeasible to run the baseline method on such size of samples.

**Network sensitivity**. Figure 3 shows Type I and Type II errors for two network models. The x-axis represents the corresponding parameter values for each model. We observe that increased parameter values exhibit higher Type II errors in general for Barabási-Albert model but not for Erdős-Rényi. A possible reason is that Barabási-Albert exhibits a more skewed degree distribution compared to Erdős-Rényi. Note that the increased parameter values indicate higher density of the network. We expect Erdős-Rényi to show a similar trend if the edge probability is further increased. NIRD outperforms KRCIT in terms of Type II error (except in Figures 3c and 3f which is due to poor calibration of the baseline method) irrespective of network density. Type II error is reduced as high as $65\%$ for Erdős-Rényi model with edge probability 0.025 (Figure 3e). Moreover, Type I error

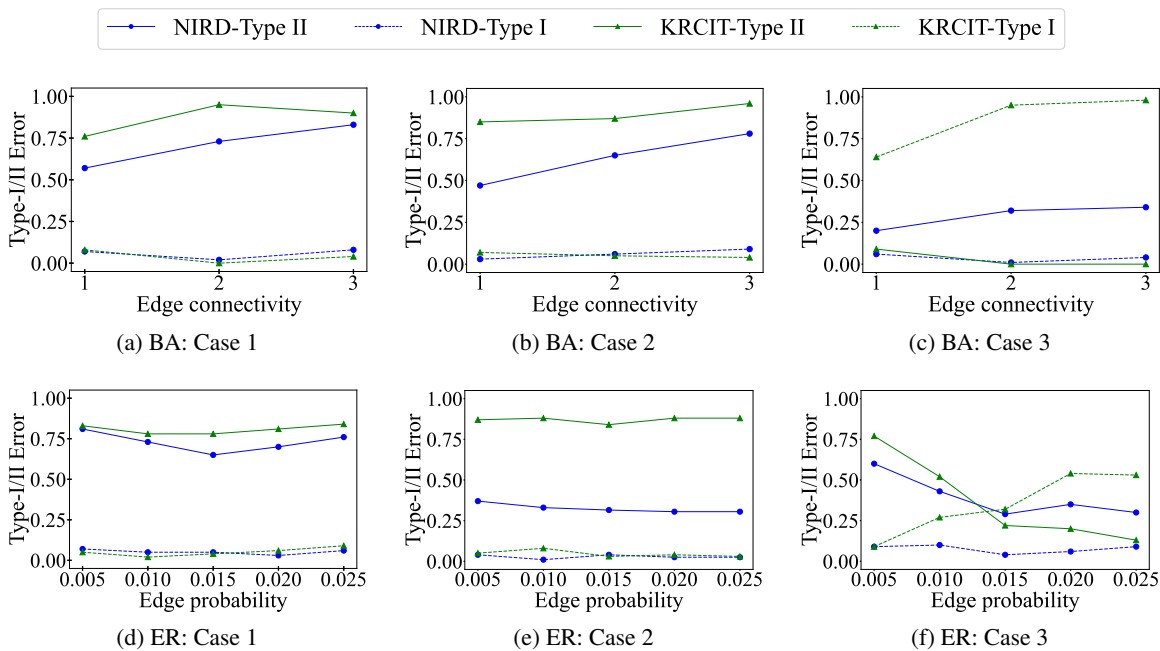

Figure 3: Impact of network parameters on Type I/II errors.

for NIRD is consistent whereas KRCIT suffers in case 2 (Figures 3c, 3f).

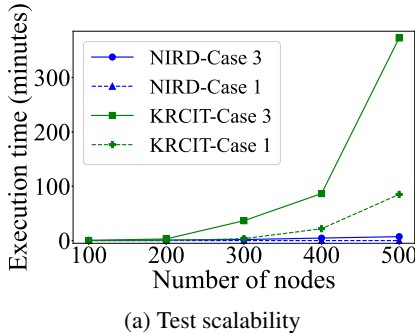

(a) Test scalability

Figure 4: Impact of sample size on execution time.

**Scalability**. Figure 4a shows execution time in minutes (y-axis) for both marginal (case 1) and conditional (case 3) independence test for different network sizes in terms of number of nodes (x-axis). The solid and dashed lines represent the conditional and marginal test result respectively. KRCIT exhibits an exponential complexity whereas NIRD shows much less sensitivity to network size. This is expected given the complexity of the corresponding algorithms.

## 6 CONCLUSION

In this work we examine the problem of defining and measuring statistical dependence in relational data. We propose NIRD, a consistent, non-parametric test for detecting relational dependence that improves state-of-the-art relational dependence testing by capturing a wide range of possible relational dependencies. Moreover, we introduce an approximate method that makes NIRD scalable to larger networks. We evaluate the effectiveness of our method across diverse relational settings and find that our proposed test exhibits significantly less sensitivity to network properties and dependence types. Our work paves the way for a number of promising future research directions, from testing for social influence to causal structure learning from relational data.

## 7 ACKNOWLEDGMENTS

This material is based on research sponsored in part by NSF under grant No. 2047899, and DARPA under contract number HR001121C0168.

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
