# OpenReview forum: "Non-Parametric Inference of Relational Dependence"
_auai.org/UAI/2022/Conference — UAI 2022 Poster_

### Official Review · Reviewer_c4mJ · 2022-04-05

**Q2(1) Originality/Novelty:** 2
**Q2(2) Significance/Impact:** 2
**Q2(3) Correctness/Technical Quality:** 3
**Q2(6) Clarity Of Writing:** 3
**Q6 Overall Score:** 6
**Q8 Confidence In Your Score:** 2

**Q1 Summary And Contributions:**

The authors propose to use a slightly modified version of HSIC for testing relational independence using non-i.i.d. observational data.

**Q2 Assessment Of The Paper:**

More detailed information regarding each of these aspects is given below:

**Q2(4) Quality Of Experiments (Optional):**

3: Good: The experimental evaluation is adequate, and the results convincingly support the main claims.

**Q2(5) Reproducibility:**

3: Good: Key resources (e.g., proofs, code, data) are available and key details (e.g., proofs, experimental setup) are sufficiently well-described for competent researchers to confidently reproduce the main results.

**Q3 Main Strengths:**

The paper is well written with a detailed introduction and a clearly defined problem setup.

The proposed method seems to perform well in numerical simulations and the authors provide a consistency result under reasonable assumptions.

**Q4 Main Weakness:**

I'm not completely sure about the novelty of the paper.

**Q5 Detailed Comments To The Authors:**

I am a bit confused about the notation after Defintion 1. $\sigma_X(v)$ corresponds to attribute X of neighbors of v. In the next sentence you use the same notation to show the special case of a propositional variable.

**Q7 Justification For Your Score:**

An interesting and well-presented application of a (slightly) modified version of the HSIC.

**Q9 Complying With Reviewing Instructions:**

1: Yes.

---

### Official Review · Reviewer_okDa · 2022-04-12

**Q2(1) Originality/Novelty:** 3
**Q2(2) Significance/Impact:** 3
**Q2(3) Correctness/Technical Quality:** 3
**Q2(6) Clarity Of Writing:** 2
**Q6 Overall Score:** 4
**Q8 Confidence In Your Score:** 2

**Q1 Summary And Contributions:**

The authors propose a kernel-based conditional independence test for settings with relational variables. They show that their test statistic is consistent, and study the performance (in terms of level and power) of the proposed test in simulated or semi-synthetic data.

**Q2 Assessment Of The Paper:**

More detailed information regarding each of these aspects is given below:

**Q2(4) Quality Of Experiments (Optional):**

3: Good: The experimental evaluation is adequate, and the results convincingly support the main claims.

**Q2(5) Reproducibility:**

3: Good: Key resources (e.g., proofs, code, data) are available and key details (e.g., proofs, experimental setup) are sufficiently well-described for competent researchers to confidently reproduce the main results.

**Q3 Main Strengths:**

The paper tackles an interesting and important problem. Studying independence relations or causal relations in relational databases and other non-iid settings is difficult so new methods in this area are very welcome.

The authors are relatively clear about their assumptions and show a clear theoretical guarantee for their test statistic (consistency).



**Q4 Main Weakness:**

There are some points of exposition that are not really clear, especially definitions that are difficult to parse/follow. (Some examples below.)

The consistency result shows a convergence in distribution, but there is no rate or uniformity guarantee, which is typically required for asymptotically valid confidence intervals and guarantees about power. In other words, though the test statistic converges to some desirable target in the limit as n goes to infinity, there's no sense of how quickly this converges, how large a sample size is needed, or if any finite large sample size is sufficient for an guarantee on the error.

The figures for the experimental results are hard to read and interpret. The lines denoting Type I error are basically uninformative because they are small and it not clear if they are above 0.05: is level controlled at the nominal 0.05 level or not? It seems that in some settings the Type I error for the proposed test varies, hard to tell what's going on.

**Q5 Detailed Comments To The Authors:**

- The first (ordinary language) definition of relational variable is unclear. "A relational variable is a set of random variables" (why a set??) "that belong to instances related to an instance of interest" (what does this mean? related how?). This could be made much more clear and formally precise. In general, for readers unfamiliar with the relational setting, more clear non-technical definitions and explanations would strengthen the paper.

- Definitions 2 and 3 depart from Daudin's in a way, and I'm not sure if this is just a mistake. Daudin's characterization is that variables are conditionally independent iff *for all smooth functions* the cross covariance vanishes. The definition in the paper has *for any smooth functions* before the "if and only if" and it makes it sound as if conditional independence depends on the chosen functions, but the point is quantifying over all functions. This should be clarified and fixed if it is indeed a mistake, though I'm not sure. I also have a question here of whether there is a corresponding distribution-level definition of independence. In the non-relational case, Daudin's result is that P(X, Y | Z ) = P(X | Z) P(Y | Z) is equivalent to this characterization by covariance across functions. Is there something similar here to in terms of distributions over the (relational) random variables involved?

- Theorem 1 and Corollary 1 are about different test statistics in some way, in one case between two propositional variables and in the other case a prop variable + relational variable. But in both statements the test statistic is just written HSIC_n which is a bit confusing and hides the difference here. This could be clarified with some specific notation to distinguish these settings.

- There is no characterization of the null distribution of the test statistic. The authors briefly mention use of permuted boot strap for calculating p-values. But is it clear that the bootstrap is valid in this setting? If the test statistic is complicated and based on non-iid data, the standard results for the bootstrap break down. Perhaps the theoretical results on V-statistics obviate this, but this issue could be clarified and elaborated. This is also related to the issue about the pointwise consistency result mentioned above -- a stronger result such that for example sqrt(n)*(HSIC_n - HSIC_pop) \to N(0,V) (or perhaps another limiting distribution) for some characterized variance would be much stronger and then wouldn't require use of the bootstrap.

- Missing a parentheses in Cov in Def 4.

- Sec 5.3 says that the experiments use standard uniform and normal noise. Problems are quite a bit easier if the noise is uniform and shares the same variance -- what if the variance of the noise varied from run to run?


**Q7 Justification For Your Score:**

The paper is interesting but could be strengthened in various ways, as outlined in Q4 and Q5.

I am willing to revise my score if the authors can address the above issues.

**Q9 Complying With Reviewing Instructions:**

1: Yes.

---

### Official Review · Reviewer_FEMd · 2022-04-12

**Q2(1) Originality/Novelty:** 3
**Q2(2) Significance/Impact:** 3
**Q2(3) Correctness/Technical Quality:** 3
**Q2(6) Clarity Of Writing:** 4
**Q6 Overall Score:** 8
**Q8 Confidence In Your Score:** 3

**Q1 Summary And Contributions:**

The authors propose a method to estimate independence a a relational dataset, Nonparametric inference of relational dependence (NIRD). Additionally they provide an approximation strategy to scale to large dataset sizes.

**Q2 Assessment Of The Paper:**

More detailed information regarding each of these aspects is given below:

**Q2(4) Quality Of Experiments (Optional):**

4: Excellent: The experimental evaluation is comprehensive and the results are compelling.

**Q2(5) Reproducibility:**

4: Excellent: Key resources (e.g., proofs, code, data) are available and key details (e.g., proof sketches, experimental setup) are comprehensively described for competent researchers to confidently and easily reproduce the main results.

**Q3 Main Strengths:**

- I enjoyed reading this paper.
- Interesting and relevant problem domain (relational and non-iid data).
- Good choice of datasets.
- Interesting to also provide approximations for large problems. This makes the paper more applicable.

**Q4 Main Weakness:**

- The paper is quite dense at times making it a bit tedious to read.
- It would have been interesting to see the effect of this test on actual structure (causal) learning. But I understand that this is beyond the scope of this work.

**Q5 Detailed Comments To The Authors:**

- "which is restricted to immediate neighbors for ease of exposition." What do you mean with ‘ease of exposition’. There are more implications to go beyond direct neighbors, no?
- More a question: Since A2 mentions that the adjacency matrix is symmetric, is there a link to lifted inference for relational domains?

**Q7 Justification For Your Score:**

Relevant, well written addition to the field of uncertainty.

**Q9 Complying With Reviewing Instructions:**

1: Yes.

---

### Official Review · Reviewer_wPmK · 2022-04-13

**Q2(1) Originality/Novelty:** 2
**Q2(2) Significance/Impact:** 3
**Q2(3) Correctness/Technical Quality:** 3
**Q2(6) Clarity Of Writing:** 2
**Q6 Overall Score:** 7
**Q8 Confidence In Your Score:** 3

**Q1 Summary And Contributions:**

This paper proposes a new (conditional) independence test for relational data that does not rely on a specific aggregate function for characterising relational dependence. This is achieved via the consideration of a kernel mean embedding which aggregates information from neighbouring nodes and which can be efficiently approximated by random Fourier features.

**Q2 Assessment Of The Paper:**

More detailed information regarding each of these aspects is given below:

**Q2(4) Quality Of Experiments (Optional):**

2: Fair: The experimental evaluation is weak: important baselines are missing, or the results do not adequately support the main claims.

**Q2(5) Reproducibility:**

2: Fair: Key resources (e.g., proofs, code, data) are unavailable but key details (e.g., proof sketches, experimental setup) are sufficiently well-described for an expert to confidently reproduce the main results.

**Q3 Main Strengths:**

The approach proposed here is definitely interesting and well grounded theoretically. The (in)dependence test proposed is likely to be used when dealing with relational variables.

**Q4 Main Weakness:**

The paper lacks clarity in some places. For example, if the marginal independence test is clear (Section 4.2), its conditional counterpart is somewhat sill obscure to me: can you better explain the motivation for using kernel ridge regression? what is the final form of the test?

Some of the choices made in the experiments are not justified. For example, the three cases retained all imply that the null hypothesis should always be rejected. It is important to understand the behaviour of the proposed test in situations where the null hypothesis should not be rejected. I strongly encourage the authors to use a collider structure in addition to the structures currently used.

**Q5 Detailed Comments To The Authors:**

In Section 4.3, the kernel ridge regression \beta should depend on x and z.

\delta is defined but not used in Definition 4. Do you in fact mean that it should be $\delta(i,j) \le r$ rather than $j - i \le r$? Or rather that \delta should not be introduced in Definition 4? The first paragraph of 4.4.2 is also misleading as you state that to extend the framework of Dedecker et al. to relational variables, one needs to replace the usual definition of distance to the shortest path distance (yet the latter is introduced in Definition 4).

As mentioned previously, it is important to consider cases in your experiments where the variables are conditionally dependent. As a minor remark, Figure 1 is not readable when printed (not mentioning the fact that the colours chosen are almost the same when printing in black and white).

**Q7 Justification For Your Score:**

This is an interesting study the results of which needs however to be strengthened before publication.

The responses provided by the authors on the issues I had are both clear and convincing. The authors state that these issues will be addressed in the final version of the paper.

**Q9 Complying With Reviewing Instructions:**

1: Yes.

---

### Decision · Program_Chairs · 2022-05-15

**Decision:**

Accept (Poster)

**Comment:**

Meta Review: The paper deals with an interesting problem. The main contribution is a new conditional independence test for relational data. The reviewers largely agree that the paper contributes new ideas, and the results are likely to have an impact on the field.

The main issues raised by the reviewers have been addressed in the authors' rebuttal. Please address these issues in the final version of the paper accordingly.